# Determination of Solidification of Rigidity Point Temperature Using a New Method

**Ester Villanueva** [1],*![ID], **Iban Vicario** [1]![ID], **Jon Mikel Sánchez** [1]![ID] and **Ignacio Crespo** [2]

[1] TECNALIA, Basque Research and Technology Alliance (BRTA), Astondo Bidea, Edificio 700, 48160 Bizkaia, Spain; iban.vicario@tecnalia.com (I.V.); jonmikel.sanchez@tecnalia.com (J.M.S.)

[2] TECNALIA, Basque Research and Technology Alliance (BRTA), Mikeletegi Pasealekua 2, 20009 Donostia-San Sebastián, Spain; ignacio.crespo@tecnalia.com

* Correspondence: ester.villanueva@tecnalia.com; Tel.: +34-943-005-511



**Featured Application: Increase the accuracy of solidification software for aluminum alloys.**

**Abstract:** This work aims to calculate the rigidity point temperature of aluminum alloys by three new methods and compare them with currently employed methods. The influence of major and minor alloying elements over the rigidity point temperature is also discussed. Until now it has been difficult to determine the exact temperature of the rigidity point, since small variations in the data obtained give variable results, making it difficult to automate the process with high accuracy. In this work we suggested three new mathematic methods based on the calculation of higher order derivatives of (dT/dt) with respect to time or temperature compared to those currently employed. A design of experiments based on the Taguchi method was employed to compare the effect of the major and minor alloying elements for the AlSi$_{10}$Mg alloy, and to evaluate the accuracy of each developed method. Therefore, these systems will allow better automation of rigidity point temperature (RPT) determination, which is one of the most important solidification parameters for solidification simulators. The importance of the correct determination of this parameter lies in its relation to quality problems related to solidification, such as hot tearing. If the RPT presents very low-temperature values, the aluminum casting will be more sensitive to hot tearing, promoting the presence of cracks during the solidification process. This is why it is so important to correctly determine the temperature of the RPT. An adequate design of chemical composition by applying the methodology and the novel methods proposed in this work, and also the optimization of process parameters of the whole casting process with the help of the integrated computational modeling, will certainly help to decrease any internal defective by predicting one of the most important defects present in the aluminum industry.

**Keywords:** rigidity point temperature; cast metal alloys; aluminum; advanced thermal analysis; microstructure

---

## 1. Introduction

Today the latest trends in the automotive industry for the manufacture of aluminum castings are based on the use of the High Pressure Die Casting (HPDC) process, especially for structural parts that are manufactured through vacuum assisted HPDC. Apart from the most employed AlSi$_9$Cu$_3$ alloy, which is employed for standard injected parts, AlSi$_{10}$Mg is the most commonly selected alloy for producing aluminum structural parts. This aluminum alloy combines high ductility values with a good crash performance [1].

A good definition of solidification characteristics of a specific aluminum alloy is very important for the accuracy of casting simulation results. Solidification of an aluminum alloy starts when in the molten

metal small crystal nuclei begin to form, producing an undercooling of the melt. The temperature of the metal at this point is called the liquidus temperature. After this point, the metal continues to decrease its temperature, promoting the precipitation of $\alpha$-Al phase dendrites. The growth of dendrites involves two main processes: first, dendrites grow forward generating primary dendrites and second, dendrites grow sideways generating secondary dendrite arms until a primary dendrite meets another primary dendrite, forming a primary dendritic network. This point is called the dendrite coherency point (DCP), with its correspondent temperature and solid fraction. A subsequent growth in the preferred crystallographic direction of secondary and even tertiary the branches follows the primary dendritic network formation, with a coarsening of the arms of the secondary dendrites when they touch each other but without being able to support any tension load.

Continuing with the cooling of the alloys, there is a moment when the coalescence ends. The temperature at this point, is known as the rigidity point temperature (RPT). The rigidity point (RP) is also called the mechanical coherence point or maximum packing fraction point. At the RP, solid bridges start to form between the solid particles and the structure becomes sufficiently coalesced and rigid to sustain considerable tensile strains and stresses with solid particles moving past each other due to the lubricating effects of the liquid film between them [2]. Interdendritic flow is restricted and should be maintained by the metastatic pressure of the feeding system or by the negative pressure caused by volume deficit, which is known as shrinkage porosity, and by capillarity forces.

Bust feeding of the casting occurs between the rigidity and solidus temperature, due to the effect of the application of an external pressure which punctures the skin of the casting exceeding the internal pressure of the casting, and finally when there is a solid feed due to surface sinking of the solidifying casting surface [3].

Adequate control of the total alloy composition is crucial in obtaining quality castings from AlSi alloys, as both major and minor alloying elements have important effects on the solidification process [4]. However, there is a lack of knowledge about how the different minor alloying elements can act over the RPT values in the AlSi$_{10}$Mg alloy, when combined with different percentages of the major and minor alloying elements and the effect of the calculation method employed to determine them. The development of the strength in the castings and the defect formation during the casting process are related to RPT. So, the analysis of the solidification region between dendrite coherency and RP is crucial in determining the effect of the alloying elements on the formation of casting defects, such as shrinkage porosity, macro-segregation, and hot tearing, to improve simulation and casting processes [5].

Thermal analysis (TA) is a widely employed system for quality control in aluminum casting plants. The obtained information can be employed in the subsequent secondary processes of the obtained cast parts. For example, the machining behavior of cast parts is very much related to the presence and quantity of the different intermetallics that can be formed during the solidification process of the alloy, promoting differences in the machinability in comparison with other manufacturing processes [6]. TA techniques can predict the presence and quantity of precipitated intermetallics, depending on the alloy composition.

In TA, plotting temperature versus time, a curve is obtained that with its derivatives is used to characterize the solidification path of the alloy [7–10]. To obtain the cooling curve, liquid aluminum is preheated approximately to 100 °C above its liquidus temperature, and then it is poured into a defined thermal analysis cup made from ceramic, steel, graphite, or sand. One, two, or more thermocouples are located in the analysis cup connected to data loggers, plotting the cooling curves. On observing only the cooling curve it is not possible to detect all the thermal events, especially when they are too weak. They are only detected using first and/or consecutive derivative curves of the cooling curve.

There are four main processes for the determination of RPT temperature. The first, the mechanical-rheological method, is based on the continuous recording of the torque required to produce the rotation of a disc or paddle in a liquid metal [11,12], until the dendrite structure becomes mechanically rigid, stopping the rotation of the impeller. The temperature at which this occurs is defined as the RPT.

The second is based on the in-situ study of neutron diffraction during casting. However, this system needs a large detector to obtain enough resolution of the diffraction peak in the semi-solid state [13].

The third method corresponds to the two thermocouples thermal analysis technique [14] to determine the temperature data at the center ($T_c$) and at a nearby inner wall ($T_w$) with two thermocouples. The RPT is determined with the second minimum on the $\Delta T$ versus time curve ($\Delta T$ = Tw − Tc) and its projection on the Tc cooling curve. Due to the difference in the thermal conductivity in the solid and liquid phases, there is a minimum on the $\Delta T$ versus time curve.

There are other methods that are based on employing only one thermocouple to decrease costs and to increase productivity of the data analysis. However, there is no work in which this method has been applied to aluminum alloys.

It seems that the application of these methods is not suitable when thermal signal of the cooling curve is weak and also in the case of hypereutectic aluminum alloys, which are studied in the following section. It has been demonstrated that the use of a derivative beyond the first derivative could extend the use of thermal analysis techniques allowing new properties related to the solidification process to be obtained.

This work suggests that the application of Taguchi techniques to obtain new regressions models based on alloy composition to determine the rigidity point for $AlSi_{10}Mg$ alloy could be a more precise and cheaper way through the advanced thermal analysis techniques. The Taguchi method has been quite often applied in the improvement and control of industrial processes in different sectors, no application of the Taguchi method has been observed in the literature. In addition, this method could also be used to analyze other types of aluminum alloys.

In the present study, a design of experiments based on the Taguchi method was employed to compare the effect of the major and minor alloying elements over the obtained RPT values, and to evaluate the accuracy of every studied method for $AlSi_{10}Mg$ alloys by employing only one thermocouple to register the solidification.

## 2. Materials and Methods

In the present work, three different methods based on the study of the effect of 12 major, minor, and trace elements in the RPT values applying the Taguchi methodology are presented. Two orthogonal matrices were studied, an L16 orthogonal matrix and a modified L8. L16 matrix incorporating two levels of every alloying element (maximum and minimum percentage of every alloying element). The L8 matrix employs intermediate values of the alloying elements, between the defined maximum and minimum. In total, according to the defined L16 and L8 matrices, 24 different alloys were studied varying the concentration of the selected alloying elements. Once the design of the tests was completely defined, each alloy was melted in an electrical resistance furnace model Heerbo72 (Morgan, Berkatal, Germany).

The base alloy selected for the study corresponds to the alloy most commonly used in the manufacture of HPDC structural parts, with different percentages of ferroalloys added to the melt to obtain the desired compositions. No master alloys for grain refining or silicon modification were added to the melts. The $AlSi_{10}Mg$ alloy was prepared according to the EN AC-43.400 standard, included in the EN 1706:2010 standard.

All alloys were poured into commercial sand cups at temperatures of approximately 700 °C. Samples taken from the melt using a commercial probe inserted into the melt were analyzed by mass spectrometry in the model SPECTROMAXX (Spectro, Kleve, Germany). The compositions obtained are summarized in Table 1.

To obtain the TA curves, alloy samples of approximately 300 g ± 10 g were poured into calibrated sand cups. Values were recorded for temperatures between 630 and 400 °C. A high-speed National Instruments Data Acquisition System (Field logger Novus) linked to a personal computer was used to

collect temperature data every 1 s. Each TA test was done at least three times with a cooling rate of about 3 °C/s.

**Table 1.** Chemical composition of studied alloys (mass %).

| Ref. | Si | Mg | Fe | Cu | Ni | Cr | Mn | Ti | Zn | Pb | Sn | Sr |
|------|-----|-----|-----|------|-----|------|------|------|------|-----|-------|------|
| 1 | 8.0 | 0.2 | 0.3 | 0 | 0 | 0 | 0.2 | 0 | 0 | 0 | 0.003 | 0 |
| 2 | 8.4 | 0.1 | 0.9 | 0.2 | 0.2 | 0.1 | 0.2 | 0.2 | 0.2 | 0.2 | 0.066 | 0.05 |
| 3 | 8.7 | 0.1 | 0.3 | 0 | 0 | 0 | 0.2 | 0.2 | 0.3 | 0.3 | 0.039 | 0.01 |
| 4 | 8.9 | 0.2 | 0.9 | 0.2 | 0.2 | 0.2 | 0.2 | 0.1 | 0.2 | 0.2 | 0.073 | 0.01 |
| 5 | 9.0 | 0.3 | 0.4 | 0 | 0 | 0 | 0.3 | 0 | 0 | 0 | 0.002 | 0.02 |
| 6 | 9.0 | 0.4 | 1.1 | 0.3 | 0.2 | 0.1 | 0.8 | 0.2 | 0.1 | 0.2 | 0.019 | 0.05 |
| 7 | 9.3 | 0.6 | 0.7 | 0.1 | 0 | 0.1 | 0.5 | 0 | 0.2 | 0 | 0.002 | 0.01 |
| 8 | 9.8 | 0.7 | 0.3 | 0 | 0.2 | 0.2 | 0.7 | 0.1 | 0.4 | 0.1 | 0.064 | 0.06 |
| 9 | 10.0 | 0.7 | 0.3 | 0 | 0.2 | 0.2 | 0.7 | 0 | 0 | 0 | 0.002 | 0.06 |
| 10 | 10.2 | 0.3 | 0.4 | 0.05 | 0 | 0.1 | 0.3 | 0 | 0.08 | 0 | 0.002 | 0.01 |
| 11 | 10.3 | 0.3 | 0.5 | 0.09 | 0 | 0.11 | 0.35 | 0.01 | 0.01 | 0 | 0.002 | 0.01 |
| 12 | 10.4 | 0.3 | 0.5 | 0.11 | 0 | 0.1 | 0.4 | 0 | 0.01 | 0 | 0.002 | 0.01 |
| 13 | 10.5 | 0.5 | 0.3 | 0.2 | 0.2 | 0 | 0 | 0.2 | 0.2 | 0.3 | 0.026 | 0.05 |
| 14 | 10.6 | 0.6 | 0.4 | 0.05 | 0 | 0.07 | 0.33 | 0.02 | 0.1 | 0 | 0.001 | 0.01 |
| 15 | 10.7 | 0.4 | 0.6 | 0.1 | 0 | 0.09 | 0.38 | 0.02 | 0.1 | 0 | 0.002 | 0.01 |
| 16 | 10.8 | 0.5 | 0.5 | 0.05 | 0 | 0.06 | 0.34 | 0.03 | 0.1 | 0 | 0.002 | 0.01 |
| 17 | 10.9 | 0.4 | 0.5 | 0.1 | 0 | 0.11 | 0.47 | 0.01 | 0.02 | 0 | 0.005 | 0.01 |
| 18 | 11.4 | 0.4 | 1 | 0.27 | 0.3 | 0.1 | 0.7 | 0.3 | 0.09 | 0.3 | 0.026 | 0.04 |
| 19 | 11.5 | 0.4 | 0.9 | 0.4 | 0 | 0.1 | 0.7 | 0.2 | 0.2 | 0.2 | 0.04 | 0.05 |
| 20 | 11.6 | 0.5 | 0.8 | 0.2 | 0 | 0.2 | 0.7 | 0 | 0.2 | 0.2 | 0.003 | 0.01 |
| 21 | 11.6 | 0.5 | 1 | 0.1 | 0.1 | 0.2 | 0.1 | 0.3 | 0.13 | 0.1 | 0.033 | 0.01 |
| 22 | 11.7 | 0.4 | 0.6 | 0.07 | 0 | 0.08 | 0.44 | 0.02 | 0.04 | 0 | 0.002 | 0.01 |
| 23 | 11.7 | 0.6 | 0.3 | 0.2 | 0.2 | 0 | 0.3 | 0.2 | 0 | 0.1 | 0.032 | 0.02 |
| 24 | 11.8 | 0.5 | 1 | 0.11 | 0.1 | 0.1 | 0.1 | 0.1 | 0.18 | 0.1 | 0.046 | 0.02 |
| 25 | 12.0 | 0.3 | 0.8 | 0.13 | 0.2 | 0 | 0.5 | 0.1 | 0.02 | 0.2 | 0.055 | 0.03 |

First, the methods described in the literature were applied and then the results obtained compared with the recently proposed methods for AlSi$_{10}$Mg alloys. Figure 1 shows the first method based on the determination of the second lowest point in the second derivative [11]. Subsequent works showed that the thermal signal is sometimes so weak that it is difficult to determine the exact point on the second derivative curve [15].

The second method is based on the determination of the minimum of the first derivative of temperature vs. time, corresponding also to the beginning of the eutectic solidification arrest [16,17]. This experimental work demonstrated that this method is only available for hypereutectic aluminum alloys. Nevertheless, it is questionable where the precise point is for hypoeutectic cases as presented in Figure 2. In this case, rather than a minimum in the first derivative of the cooling curve, there are several turning points, so it is not possible to detect the exact RPT.

These studies indicated the need for new methods, which is of course what this research is all about. Three new methods developed to determine the dendritic consistency point are proposed based on the one used previously by the authors [18,19] but using different criteria for determining this

thermal arrest. An analysis of the improvements and results obtained by applying each method is be made in the following section.

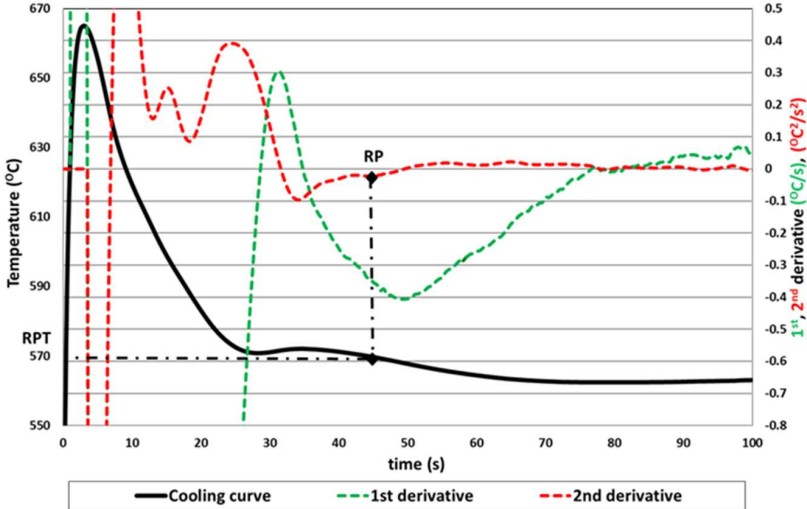

**Figure 1.** Rigidity point temperature (RPT) determination with the second minimum of the d2T/dt2 curve.

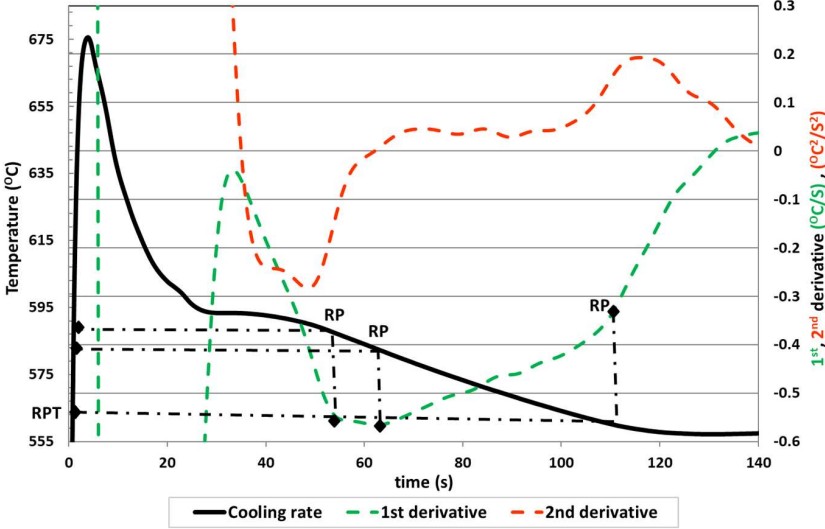

**Figure 2.** Method 2: RPT determination with the first minimum of the dT/dt curve: Hypoeutectic AlSi$_{10}$Mg alloy.

The process consisted of calculating the first and successive derivatives of the temperature versus time curve and applying the following proposed methods.

The first method proposed is based on the determination of the zero-intersection point of the second and third temperature derivatives with respect to the time curve after the maximum temperature of liquidus (Method 3). It is based on previous work by David Sparkman developed for iron alloys, in which he justified that taking into account that at this point the dendrites have finished growing together and that the eutectic begins to grow and release energy, the rigidity point is also the starting point for the arrest of eutectic solidification [18].

We can observe the first proposed method for the determination of the RPT in Figure 3 with the RP close to the minimum of the first derivative.

In Figure 2 we can observe how the determination of the exact minimum of the first derivative is not intuitive, and in the case of the proposed method we can obtain the RPT value directly.

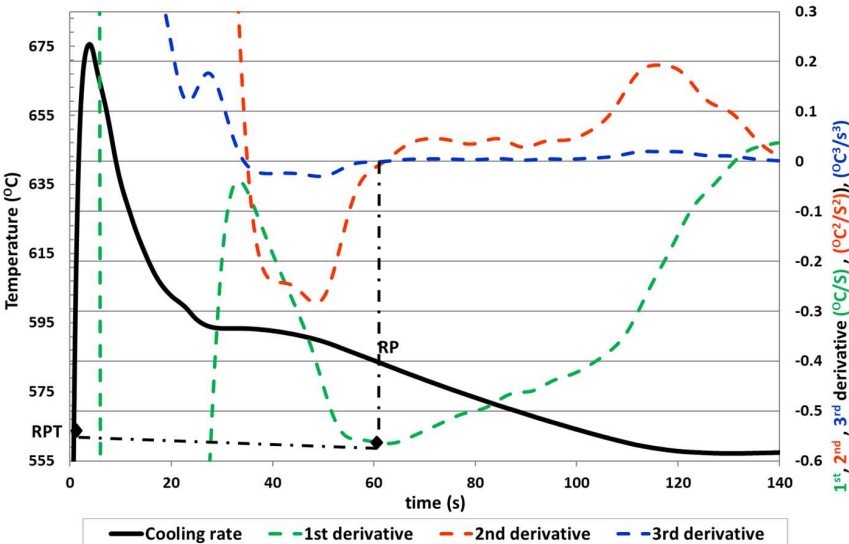

**Figure 3.** Method 3: RPT determination in the crossing point of the second and third derivative dT/dt vs. time for a hypoeutectic AlSi$_{10}$Mg alloy.

The second proposed method (Method 4 in the following Figure 4) for obtaining RPT using a single thermocouple is based on the determination of the elbow point at which the dT/dt curve suddenly deviates from the horizontal tangent in the first derivative of the temperature curve with respect to time (dT/dt) but represented vs. temperature. This method assumes the fact that the use of derivatives with respect to temperature is not influenced by the size of the thermal cup and also that an acceleration of dT/dt versus temperature corresponds to an increase in speed of the heat extraction from the sample due to the higher conduction of the heat in a solidified net. The process is based on the same procedure that Victor Anjos proposed to determine the dendrite coherence temperature point [19]. He justified for hypoeutectic ductile iron alloys that the moment where the second and third derivative curve cross the zero line after the liquidus temperature corresponds to the DCP, with the first minimum of the first derivative after the eutectic minimum temperature for eutectic iron alloys and the first minimum of the second derivative after the minimum liquidus temperature arrest for hypereutectic iron alloys.

We can observe in Figure 4 the determination of RPT in the elbow of the dT/dt curve versus time and the corresponding RPT in the sudden deviation from the horizontal tangent.

However, it is often not easy to find the exact point at which the horizontal tangent deviates, since there is not always a clear loop in the obtained curve and, consequently, it is not possible to determine the exact position of the elbow point on the dT/dt versus temperature curve.

Therefore, another method (Method 5) is proposed based on the previous one in which the determination of the RPT coincides with the point of intersection with the zero of the second and third derivative after the maximum temperature of liquidus, to obtain a more accurate rigidity temperature point.

We can observe in Figure 5 the determination of the RPT for the third proposed method.

The last step of the study was to analyze the microstructure to correlate the RPT values with the microstructure of the alloy. Several ingots of approximately 50 mm (length) × 50 mm (width) × 50 mm (thickness) were obtained for each alloy. The samples for optical microscopy (OM) were cut from these ingots and prepared according to standard metallographic procedures, by hot mounting in conductive resin, then grinding, and polishing. The microstructure and the determination of the possible metallographic phases of each sample were investigated by an optic microscope model DMI5000 M (LEICA Microsystems, Wetzlar, Germany).

A scheme of the complete method applied is shown in the Figure 6.

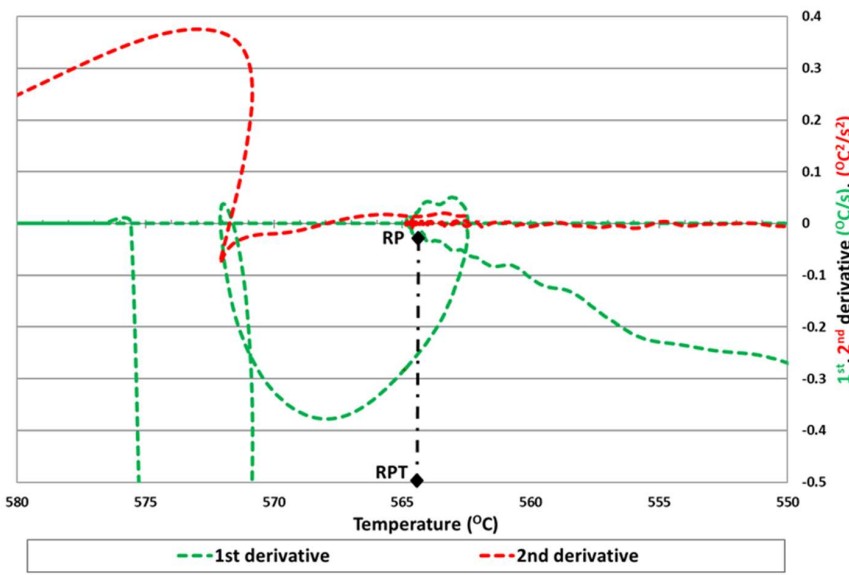

**Figure 4.** Method 4: dT/dt curve vs. T, with the RPT point in the elbow.

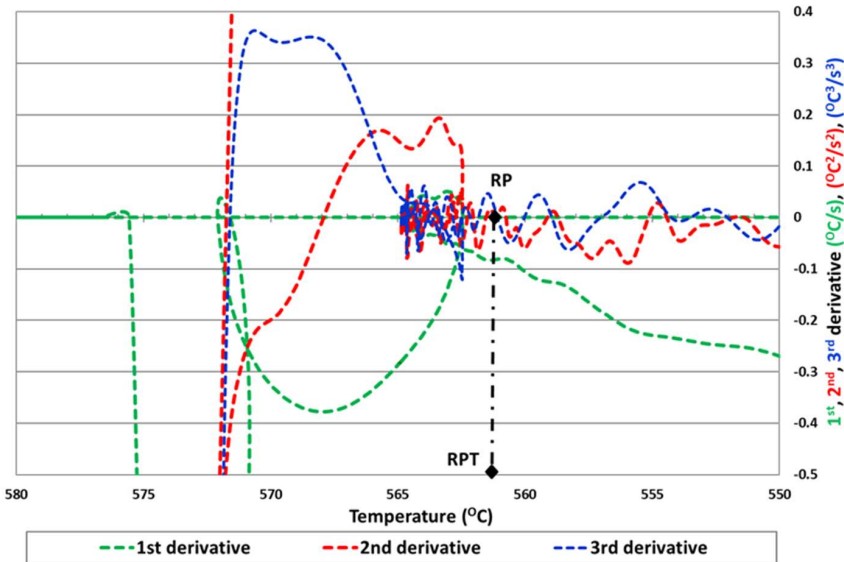

**Figure 5.** Method 5: RPT determination in the crossing point of the second and third derivative of dT/dt vs. T curve.

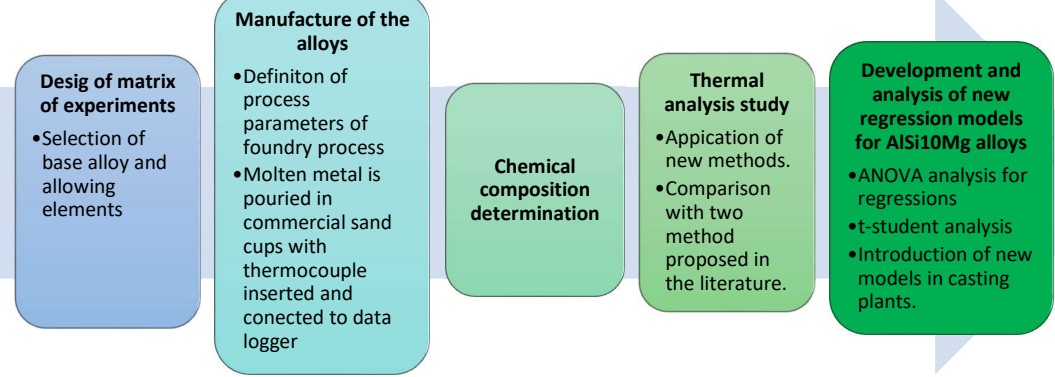

**Figure 6.** Procedure followed in this research work.

## 3. Results

From the solidification curves and applying higher order derivatives, the rigidity point temperatures (RPT) were determined with the different studied methods and the comparison with the two methods presented in the literature while considering the gaps of both methods.

In the following equations, from number 1 to 5, the linear regression equations obtained using the different methods studied for the calculation of the RPT in function of the alloy composition are shown.

**Method Equation (1)**

$$RPT(°C) = 564.48 + 0.65.Si − 8.48.Mg − 2.37.Fe + 6.09.Cu − 0.26.Ni + 3.56.Cr − 4.46.Mn − 9.01.Ti − 4.89.Zn + 1.05.Pb − 7.79.Sn − 0.002.Sr \tag{1}$$

**Method Equation (2)**

$$RPT(°C) = 564.99 + 0.59.Si − 8.28.Mg − 2.35.Fe + 5.90.Cu − 0.32.Ni + 3.42.Cr − 4.47.Mn − 8.84.Ti − 5.36.Zn + 1.37.Pb − 7.84.Sn + 1.97.Sr; r^2 = 0.77 \tag{2}$$

**Method Equation (3)**

$$RPT(°C) = 567.74 + 0.58.Si − 8.24.Mg − 2.89.Fe + 7.41.Cu − 2.04.Ni + 3.90.Cr − 4.25.Mn − 7.70.Ti − 6.24.Zn + 0.37.Pb + 5.13.Sn − 5.20.Sr \tag{3}$$

**Method Equation (4)**

$$RPT(°C) = 560.92 + 0.76.Si − 10.62.Mg − 1.42.Fe + 3.06.Cu − 1.95.Ni + 0.69.Cr − 2.01.Mn − 3.74.Ti − 0.16.Zn − 2.09.Pb − −15.03.Sn + 18.42.Sr. \tag{4}$$

**Method Equation (5)**

$$RPT(°C) = 565.96 + 0.55.Si − 7.97.Mg − 2.76.Fe + 6.50.Cu − 0.01.Ni + 3.83.Cr − 4.58.Mn − 8.20.Ti − 5.75.Zn + 1.16.Pb + 0.29.Sn − 10.33.Sr \tag{5}$$

Equations (1)–(5) show that Mg, Cu, Cr, Mn, Ti, Zn, and Sn have in general a decreasing effect on RPT values. Si, Cr and Cu are the only elements that promote an increase of RPT values. Alloying elements Mg, Fe, Ni, Mn, Ti, and Zn decreases RPT values, and the rest of the alloying elements have positive or negative values in the function of the employed method. Pb, Sn, Sr do not show a specific influence as in some methods their influence on RPT is positive and in other methods negative.

The rigidity temperature values obtained employing the different studied methods are displayed in the following Figure 7 in order to compare the differences and tendencies of the obtained values.

Methods 1, 2, 3, and 5 have very similar temperature values, while Method 4 shows lower RPT values. Furthermore, it can be appreciated that the influence of the alloy elements is clear on the values obtained with similar differences regardless of the method selected. Small changes in the composition of the alloy promote important changes in the RPT values. There is up to approximately 10 °C of difference in the RPT values with the same method depending on the real alloy composition.

Table 2 shows the accuracy of the RPT values obtained for each method, with the linear regression coefficient ($r^2$) and the standard error ($S_{ey}$).

Method 1 and Method 2 showed similar coefficient of determination values compared to Method 3 and Method 4, with values between 0.77 and 0.78. Method 5 using the plotting of the derivatives vs. temperature parameter is the one with more precision, with 0.82. The use of higher order derivatives which made easier the determination of the RPT and the plotting of the dT/dt curve vs. T, that is independent of the sand cup size provides better results in Method 5. A value of the coefficient of determination between 0.5 and 0.8 is classified as a regular correlation and a value from 0.8 to 0.9 is

classified as a good correlation. We can observe that Method 5 is the only one characterized as a good correlation, with the rest of the methods very close in obtaining this classification.

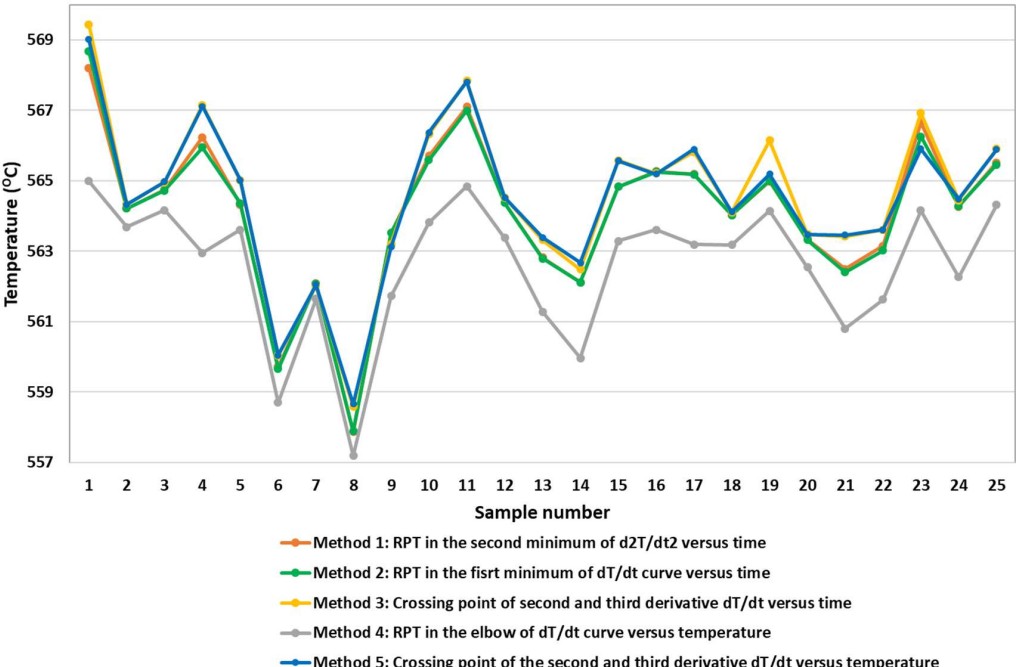

**Figure 7.** Comparison of the RPT values for each alloy with the methods studied.

**Table 2.** Description of the determination of the rigidity point temperature by every method and the corresponding coefficients of determination, standard error, and maximum and minimum deviation.

| | Description of the Determination of the RPT | Coefficient of Determination $R^2$ | Standard Error | Maximum and Minimum Deviation |
|---|---|---|---|---|
| Method 1 | Second Minimum of $d^2T/dt^2$ curve versus time | 0.77 | 1.48 °C | −0.40/+ 0.30% |
| Method 2 | First minimum of dT/dt curve versus time | 0.77 | 1.5 °C | −0.39/+ 0.36% |
| Method 3 | Crossing point of the second and third derivative dT/dt versus time | 0.78 | 1.55 °C | −0.48/+ 0.39% |
| Method 4 | Sudden deviation from the horizontal tangent in the elbow area of dT/dt curve versus temperature | 0.77 | 1.29 °C | −0.35%/+ 0.29% |
| Method 5 | Second and third derivative of dT/dt versus temperature | 0.82 | 1.35 °C | −0.44%/+ 0.32% |

The standard error is similar for Methods 4 and 5, and slightly higher in Methods 1, 2, and 3. A difference of about 1.4 °C in the RPT with deviation percentages of about 0.4% are acceptable when the approximative RPT temperature is about 565 °C.

In order to know the significant effect of each alloy element on the RPT, the values of the Student's t-distribution (t) were used. In our case, if the t value is >2.17, the selected alloy element will have a significant influence on the studied parameter. If the value of a coefficient in the RPT calculation formulas is negative, the selected alloy element will promote a decrease in the RPT value, and a positive value will promote an increase in the RPT value. In Table 3 the individual values for the t student of every alloying element are resumed.

**Table 3.** Student t values for regression models for calculation of RPT.

| Meth. | Si | Mg | Fe | Cu | Ni | Cr | Mn | Ti | Zn | Pb | Sn | Sr |
|---|---|---|---|---|---|---|---|---|---|---|---|---|
| 1 | 1.65 | 2.14 | 1.21 | 0.95 | 0.04 | 0.47 | 1.78 | 1.37 | 0.64 | 0.13 | 0.27 | 0.00 |
| 2 | 1.48 | 2.05 | 1.18 | 0.91 | 0.05 | 0.45 | 1.75 | 1.32 | 0.69 | 0.17 | 0.27 | 0.07 |
| 3 | 1.40 | 1.99 | 1.41 | 1.11 | 0.29 | 0.50 | 1.62 | 1.12 | 0.80 | 0.05 | 0.17 | 0.18 |
| 4 | 1.53 | 2.21 | 1.54 | 1.11 | 0.00 | 0.56 | 2.00 | 1.37 | 0.82 | 0.16 | 0.01 | 0.41 |
| 5 | 2.22 | 3.09 | 0.83 | 0.55 | 0.33 | 0.11 | 0.92 | 0.66 | 0.02 | 0.31 | 0.61 | 0.76 |

We can observe in Table 3 that the obtained results show that magnesium was determined as the main element impacting on the RPT values and it is the element with the highest t student coefficient and with statistical relevance. Silicon also has a statistical signification in the case of Method 5, with average values in the rest of the studied methods quite close to the limit of also having statistical signification.

In Figure 8 we can observe the representation of the Mg effect over RPT values employing the calculation of method 5. Higher Mg percentages promote smaller RPT values.

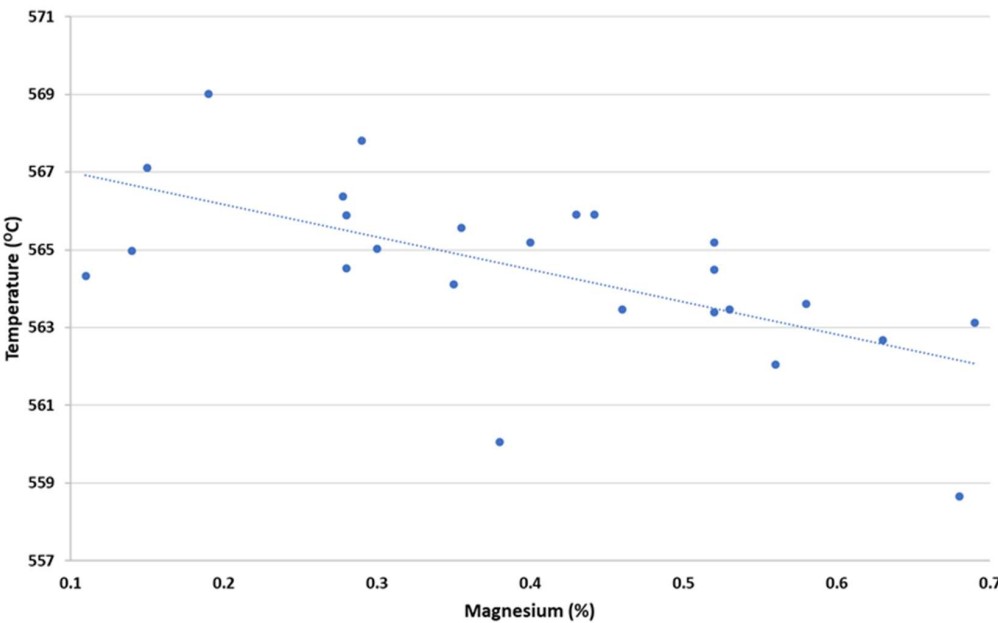

**Figure 8.** Effect of Mg percentage over RPT employing Method 5.

## 4. Discussion

In this study we compared the different methods in order to determine the best method to obtain the RPT of $AlSi_{10}Mg$ alloys.

**Method 1** sometimes is not suitable for the determination of RPT, because in the second derivation curve two minimum can be presented, as is shown in Figure 1.

**Method 2** was designed for hypereutectic alloys, and it is sometimes difficult to determine the exact point for the RP of the hypoeutectic aluminum alloys. The difficulties in determining the exact point reduces the accuracy of the calculations, and for example magnesium has no statistical signification after analyzing the t student coefficient.

**Method 3** is based on the application of higher order derivatives of the temperature curve with respect to time, with a very similar coefficient of determination in comparison to those of Method 1 and Method 2. However, neither of these methods reflects the effect of magnesium over on the RCP values in the regression models. The reason seems to be that the application of the method is not as accurate because the size of the thermal cup plotting the dT/dt curve vs. time is not considered.

**Method 4** is based on the use of a derivative curve but plotting dT/dt versus temperature, and the rest of the methods are based on the moment when the cooling rate is constant, so that all the dendrites touch each other, compared with the concept of acceleration in Method 4, where only a limited number of dendrites touch each other, which may promote an increase in the cooling rate of the sample. It is proven that temperature is a more reliable parameter because it is not influenced by the cup size. Thus, it has the smallest standard error (1.29 $^\circ$C). However, the rest of the results are very similar to those of Method 1, 2, and 3, because it is quite difficult to define the exact RPT value, as the acceleration of the cooling rate is the base to determine the exact point where the RPT starts. It is not so easy to determine the exact point where the tangent is in the elbow area of the curve.

**Method 5** is the most accurate process for determining the RPT values, with the highest coefficient of regression, combined with a small error. In addition to using higher order derivatives, it uses the dT/dt derivative curve with respect to the temperature in order to define the RPT. It is similar to Method 3 in using higher order derivatives to determine the RPT but in this case the temperature derivative curve with respect to the time is in a way similar to Method 4, thus the coefficient of regression is reduced.

By the study of student t results, we can observe that the method with the highest values is obtained for Method 5. The results obtained show a result that is consistent with previous results in the literature, since it has been determined that magnesium is the main element that influences the RPT values and is the element with the highest student t coefficient [20]. This method is the only one that indicates that silicon also has a statistical effect on the RPT values. This is in congruence with the studies of the variation of DCP temperatures as function of the alloy composition, in that it has a very similar effect on the solidification path of aluminum alloys. At the DCP there is also an increase in the cooling rate.

In general, the regression models demonstrated that when the percentage of an alloying element such as Mg, Fe, Ni, Mn, Ti, and Zn is increased, the RPT values tend to decrease. The increase of percentage of alloying elements in an Al alloy decrease the solidification process related temperature until the minimum solidification temperature, which is reached in the eutectic composition. Giraud and his team [20] observed that the coalescence solid fraction is independent of Si content and depends on Mg content for $AlSi_{10}Mg$ alloys, because a solid skeleton is developed earlier in the Mg- rich alloys, like $\pi$-$Al_8Mg_3FeSi_6$. The Fe and Ni as alloying elements can form $\beta$-$Al_5SiFe$, $Al_3Ni$, and $Al_7Cu_2Fe$ needle like structures, that they can reduce the solid particles moving to pass each other, as shown in Figure 9.

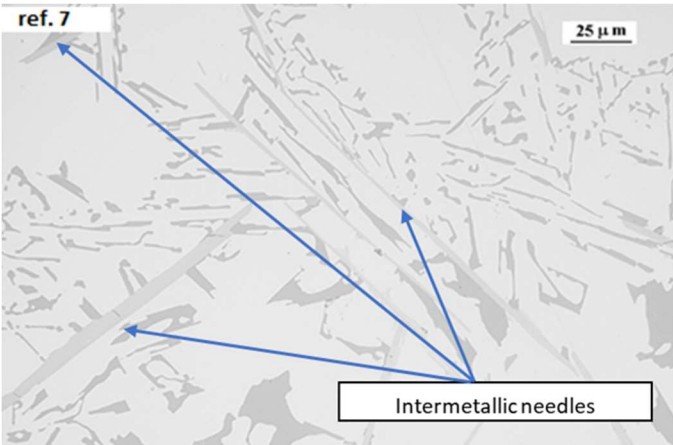

**Figure 9.** Alloy 7 micrography showing needle like structures of intermetallic containing high Fe concentrations, Ni and Cu.

Zn and Mn are normally dissolved in molten aluminum alloy inside the grains, so when combined, an increase of the wt% of Zn and Mn takes place as a more eutectic alloy. Mn also has a direct effect over $\beta$-$Al_5SiFe$, modifying it to $\alpha$-$Al_{15}Si_2(Fe, Mn)_3$ with a more compact Chinese script morphology, which is normally also precipitated at higher temperatures than the RPT. In we can observe the presence of Chinese script structures and the reduced presence of $Al_5SiFe$ and $Al_7Cu_2Fe$ needle like structures in a high manganese $AlSi_{10}Mg$ alloy.

In the case of Ti, its effect on the RPT can be explained by the refining effect on the grain size, that promotes smaller grains and increases fluidity.

The only elements that increase the RPT values are Cu, Cr, and Si (with a statistical relevance in Method 5). Cu is present in $AlSi_{10}Mg$ alloys as $Al_2Cu$ or as complex intermetallic combined with Fe and Si. $Al_2Cu$ phase is formed at the final stage of the solidification, at temperatures very close to the solidification point, with an enrichment in Cu in the remaining eutectic liquid. So, Cu precipitates do not decrease the fluidity of solid particles. The combination of Cu with Fe can also modify the beta needle like structure of iron intermetallic to skeleton like structures, as $Q$-$Al_5Mg_8Cu_2Si_6$, as observed in Figure 10. Cr addition to the alloy changes the beta iron-rich intermetallic into modified alpha phases and therefore it reduces the detrimental effect of iron and the blocking effect on the solid particles moving into the solidifying alloy [21]. An increase of Si moves an alloy from a hypoeutectic to a more eutectic alloy, increasing the RPT in the same way that occurs with the maximum and minimum eutectic nucleation temperature [8]. An increase in high Si levels in the hypoeutectic alloys increase the Al-Si eutectic temperature and therefore the RPT is also increased. Si addition increases volumetric contraction by the formation of a silicon phase, hence reducing shrinkage porosity [22].

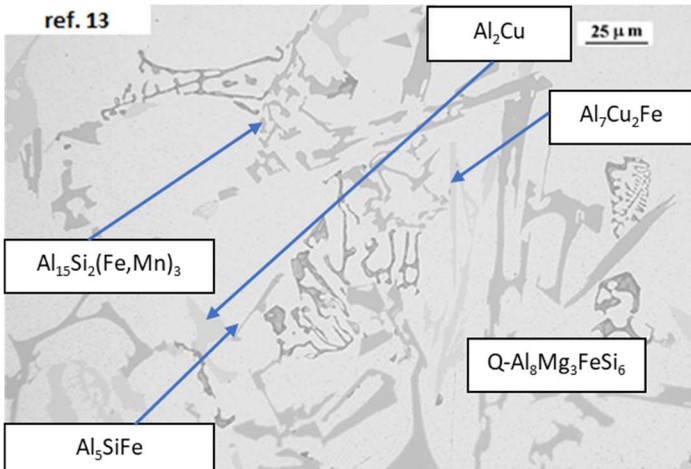

**Figure 10.** Alloy 13 micrography showing Al15Si2(Fe,Mn)3, Al2Cu, Al5SiFe, Q-Al8Mg3FeSi6 and Al7Cu2Fe structures.

Analyzing the data from Table 1 and Figure 7, we can observe that samples from number 6 to 9 have the smallest RPT values, compensating for the increase of RPT values by increasing the Si wt%. These three alloys have the highest wt% values for Mg, Fe, and Mn alloying elements, so they have a minor effect on the RPT obtained values. In the regression equation obtained from the studied methods, allowing both elements, Fe and Mn presents student t coefficient high values but without having a statistical significance in Method 5.

The rest of the alloying elements also have an influence on the RPT; however, they are not so relevant.

In general, the rest of the alloying elements are found in very small percentages so their influence using one coefficient, or another is insignificant, but the combination of all the alloying elements has an influence over the final RPT obtained values.

## 5. Conclusions

A procedure based on the Taguchi method was employed to calculate the RPT values for different alloy compositions. The obtained results presented in this work display the importance of the composition of the alloy over RPT values and the differences for RPT values obtained employing different calculation methods. The results show that the obtained equations are useful for determining with a good accuracy the RPT value of any alloy of the AlSi$_{10}$Mg family, with a good statistical correlation coefficient applying the newly developed Method 5.

Mg, Ti, Mn, Fe, and Zn are the elements that have the strongest negative influence over the RPT values and Si, Cu and Cr the positive ones. Mg and Si are the two elements with have more influence on the RPT values. In the case of elements Sn and Sr, because their content is too low they do not have an influence on the RPT. The rest of the alloying elements despite not having a statistical signification also have an influence on the final RPT values.

This paper determined that the analysis of the cooling curve taken from the temperature registration on the center of the cup with only a thermocouple can be successfully used to accurately predict the RPT.

The determination of the RPT values in Method 5 setting the point where the second derivative crosses the third derivative curve after the point related to the maximum liquidus temperature, provides an easier and more precise tool, employing the dT/dt vs. T curve, where the influence of the test cup is much less significant. These new systems based on the application of advanced thermal analysis allow a better automatization of the RPT value determination to be used with industrial TA equipment and simulation software.

More investigations with torque measurements should be done in order to determine which one has the accuracy of the proposed method for the real RPT point determination. In future works, the percentage of the solid fraction at the rigidity temperature point with the newly proposed method, combined with Calphad methodology, will be studied.

**Author Contributions:** I.V., E.V., and J.M.S. conceived and designed the experiments; I.V., J.M.S., I.C., and E.V. performed the experiments; E.V. conducted the investigation and prepared the data, E.V. and J.M.S. analyzed the data; I.V. carried out the validation; E.V. wrote the paper. All authors have read and agreed to the published version of the manuscript.

**Funding:** This research was partially funded by the Basque Government through the Etorgai Programme 2016, under the Filing Identification Number ZE-2016/00018 and the Elkartek Programme 2020, KK-2020-00047 for research, technological development and demonstration.

**Conflicts of Interest:** The authors declare no conflict of interest.

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
