# Peer review of "Determination of Solidification of Rigidity Point Temperature Using a New Method"

_applsci, doi:10.3390/app10072472_

Round 1
Reviewer 1 Report
The manuscript presents results of great importance to the professionals dealing with the casting of parts made of aluminium alloys. On the other hand these results are neither analyzed nor discussed with the precision meeting the expectations of research community interested in physical metallurgy of aluminium alloys. Discussion of the results is reduced to c.a. one page (+ two figures). The conclusions do not address the role of the elements taken into account in the context of the relevant phase diagrams. The role of Sn and Sr is not commented.
My suggestion to the Authors is to divide more clearly the manuscript into methodological issues and application to the aluminium alloys. It would be nice if the data in Table 1 are visualized. Table 2 is too hard to digest – remove. Figure 6 should be re-organized – sample number is not relevant characteristic of the alloy investigated.
Author Response
Point 1: The manuscript presents results of great importance to the professionals dealing with the casting of parts made of aluminium alloys. On the other hand these results are neither analyzed nor discussed with the precision meeting the expectations of research community interested in physical metallurgy of aluminium alloys.
Response 1: In the new manuscript in line number 266 a new table (Table 2) has been introduced which make a complete analysis of every method. Also in line 269 new description of the different methods including disadvantages has been introduced:
“Method 1 and Method 2 showed similar coefficient of determination values than Method 3 and Method 4, with values between 0.77-0.78. Method 5 using plotting the derivatives vs temperature parameter is the one with more precision, with 0.82. The use of higher order derivatives that made easier the determination of the RPT and the plotting of the dT/dt curve vs. T, which it is independent of the sand cup size provides better results in Method 5. A value of the coefficient of determination between 0.5 and 0.8 is classified as a regular and a value from 0.8 to 0.9 is classified as a good correlation. We can observe that Method 5 is the only one characterized as good correlation, with the rest of methods near to obtain this classification.”.
Also, in line number 159 and 163 new figures (figure number 1 and number 2) have been modified, to better explain the disadvantage of each method.
Point 2: Discussion of the results is reduced to c.a. one page (+ two figures).
Response 2. According to the suggestion, discussion of the results have been extended. In line number 300 following text has been added:
“In this study we compared the different Methods in order to determine the best method to obtain the RPT of AlSi10Mg alloys.
Method 1 sometimes is not suitable for the determination of RPT, because in the second derivation curve two minimum can be presented, as it is shown in Figure 1.
Method 2 was designed for hypereutectic alloys, and it’s sometimes difficult to determine the exact point for the RP of hypoeutectic aluminum alloys. The difficulties on determining the exact point reduces the accuracy of the calculations, and for example magnesium has no statistical signification after analyzing t student coefficient.
Method 3 is based on the application of higher order derivatives of temperature curve with respect to time, with a very similar coefficient of determination in comparison to those of method 1 and method 2. However, neither of these methods reflects the effect of magnesium over the RCP values in the regression models. The cause seems to be that the application of the method is not as accurate because they don’t consider the size of the thermal cup plotting the dT/dt curve vs. time.
Method 4 is based on the use of derivative curve but plotting dT/dt versus temperature, and the rest of methods are based on the moment when the cooling rate is constant, so that all the dendrites touch each other, compared with the concept of acceleration in Method 4, where a limited number of dendrites touch each other, which may promote an increase in the cooling rate of the sample. It is proven that temperature is a more reliable parameter because it is not influenced by the cup size. Thus, it has the smallest standard error (1.290C). However, the rest of results are very similar to those of Method 1, 2 and 3, because it’s quite difficult to define the exact RPT value, because the acceleration of the cooling rate is the base to determinate the exact point where the RPT starts and it’s not so easy to determinate the exact point where it’s the tangent in the elbow area of the curve.
Method 5 is the most accurate process for determining the RPT values, with the highest coefficient of regression, combined with a small error. In addition to using higher order derivatives, it uses the dT/dt derivative curve with respect to the temperature in order to define the RPT. It’s similar to method 3 in using higher order derivatives to determine the RPT but in this case the temperature derivative curve with respect to the time in a way similar to Method 4, thus the coefficient of regression is reduced.
By the study of student t results, we can observe that the method with the highest values are obtained for method 5. The results obtained show a result that is consistent with previous results in the literature, since it has been determined that magnesium is the main element that influences the RPT values and is the element with the highest student t coefficient in all the [20]. This method is the only one that indicates that silicon also has a statistical effect on the RPT values. This is in congruence with the studies of the variation of DCP temperatures in function of the alloy composition, that it has a very similar effect over the solidification path of aluminum alloys. At DCP there is also an increase in the cooling rate.”
Point 3: The conclusions do not address the role of the elements taken into account in the context of the relevant phase diagrams. The role of Sn and Sr is not commented.
Response 3: In the line number 388 it is introduced the role of Sn, Sr. The concentration for these elements is so low that hardly have influence over Rigidity Point Temperature.
Point 4: My suggestion to the Authors is to divide more clearly the manuscript into methodological issues and application to the aluminium alloys. It would be nice if the data in Table 1 are visualized. Table 2 is too hard to digest – remove. Figure 6 should be re-organized – sample number is not relevant characteristic of the alloy investigated.
Response 4: The section of Materials and method have been completed introducing the detailed description about the different methods which had seen introduced in the previous version in the section of the abstract.
And as suggested table 2 has been eliminated. Instead of providing a table with all of the values obtained for each alloy, it was thought more convenient to present a table summarizing the type of method applied and the results obtained in terms of dispersion and regression.
Figure 6 has been reorganized so that the now difference in how each method is applied is shown.
Reviewer 2 Report
The authors use three different methods to determine the rigidity point temperature (RPT) for analysis of Aluminum cast parts. Two commonly used methods are presented. Three new methods are proposed, trying to simplify the calculation process and to get a more accurate result. Based on the Taguchi methodology, the authors use these 5 different methods to determine the RPT of 24 different alloy. The paper shows that the new methods have similar regression coefficient and standard error. Also, the significance of each element is studied by introducing Student t distribution.
In general, the comparison between new methods and present methods does not directly show the advantage of the new ones. On the other hand, reasons about these new methods will allow better automation of RPT is not well explained.
In addition, Introduction about the new methods is ambiguous. It is suggested to modify the sentence for a better explanation. It will be better if the detailed description can be moved to the method section. And a brief explanation of why the new methods have its advantage should be presented.
It would be nice to show the schematic on the experiment setup, it is not so clear whether the experiment setup of the new methods is the same as the commonly used ones. In section 2, there is only one subsection 2.1, is there anything missed here? Also, Is there any advantage or disadvantage of each of the three methods?
Author Response
Point 1: The authors use three different methods to determine the rigidity point temperature (RPT) for analysis of Aluminum cast parts. Two commonly used methods are presented. Three new methods are proposed, trying to simplify the calculation process and to get a more accurate result. Based on the Taguchi methodology, the authors use these 5 different methods to determine the RPT of 24 different alloy. The paper shows that the new methods have similar regression coefficient and standard error. Also, the significance of each element is studied by introducing Student t distribution. In general, the comparison between new methods and present methods does not directly show the advantage of the new ones.
Response 1: According to the suggestion, in line number 266 a new table (number 2) collecting the main characteristics of every method is offered. Also in line 269 new description about the different methods is provided.
“Method 1 and Method 2 showed similar coefficient of determination values than Method 3 and Method 4, with values between 0.77-0.78. Method 5 using plotting the derivatives vs temperature parameter is the one with more precision, with 0.82. The use of higher order derivatives that made easier the determination of the RPT and the plotting of the dT/dt curve vs. T, which it is independent of the sand cup size provides better results in Method 5. A value of the coefficient of determination between 0.5 and 0.8 is classified as a regular and a value from 0.8 to 0.9 is classified as a good correlation. We can observe that Method 5 is the only one characterized as good correlation, with the rest of methods near to obtain this classification.
The standard error is similar for Method 4 and 5, and slightly higher in Method 1,2 and 3. A difference of about 1.4ºC in the RPT with deviation percentages about 0.4% are acceptable when the approximative RPT temperature is about 565ºC.”
And in line 163 new figure demonstrate that the method is not available for all type of AlSi10Mg alloys.
Figure 2: Method 2: RPT determination with the first Minimum of dT/dt curve: Hypoeutectic AlSi10Mg alloy
Point 2: On the other hand, reasons about these new methods will allow better automation of RPT is not well explained.
Response 2: In the line number 23 corresponding to the summary of the work developed we explained the reason why the rigidity point is so important, and the new method will allow better automation of RPT.
Point 3: In addition, Introduction about the new methods is ambiguous. It is suggested to modify the sentence for a better explanation. It will be better if the detailed description can be moved to the method section.
Response 3: As suggested all detailed description about the new methods have been moved to lines number 167 corresponding to section of Materials and Methods.
“These studies indicated the need of new methods, which is what this research is all about. Three new methods developed to determine the dendritic consistency point are proposed based on the one used previously by the authors [18][19] but using different criteria for determining this thermal arrest. An analysis of the improvements and results obtained by applying each method will be made in the following section.
The process consisted of calculating the first and successive derivatives of the temperature versus time curve and applying the following proposed methods.
The first method proposed is based on the determination of the zero-intersection point of the second and third temperature derivatives with respect to the time curve after the maximum temperature of liquidus (Method 3). It is based on previous work by David Sparkman developed for iron alloys, in which he justified that taking into account that at this point the dendrites have finished growing together and that the eutectic begins to grow and release energy, the rigidity point is also the starting point for the arrest of eutectic solidification [18].
We can observe the first proposed method for the determination of the RPT in Figure 3 with the RP close to the minimum of the first derivative.
In the figure we can observe how the determination of the exact minimum of the first derivative is not intuitive, and in the case of the proposed method we can obtain the RPT value directly.
The second proposed method (Method 4 in the following figure) for obtaining RPT using a single thermocouple is based on the determination of the elbow point at which the dT/dt curve suddenly deviates from the horizontal tangent in the first derivative of the temperature curve with respect to time (dT/dt) but represented vs temperature. This method assumes the fact that the use of derivatives with respect to temperature are not influenced by the size of the thermal cup and also that an acceleration of dT/dt versus temperature corresponds to an increase in speed of the heat extraction from the sample due to the higher conduction of the heat in a solidified net. The process is based on the same procedure that Victor Anjos proposed to determine the dendrite coherence temperature point [19]. He justified for hypoeutectic ductile iron alloys that the moment where the second and third derivative curve cross the zero line after the liquidus temperature corresponds to the DCP, with the first minimum of the first derivative after eutectic minimum temperature for eutectic iron alloys and the first minimum of the second derivative after minimum liquidus temperature arrest for hypereutectic iron alloys.
We can observe in Figure 4 the determination of RPT in the elbow of the dT/dt curve versus time and the corresponding RPT in the sudden deviation from the horizontal tangent.
However, it's often not easy to find the exact point at which the horizontal tangent deviates, since there is not always a clear loop in the obtained curve and, consequently, it’s not possible to determine the exact position of the elbow point on the dT/dt versus temperature curve.
Therefore, another method (method 5) is proposed based on the previous one in which the determination of the RPT coincides with the point of intersection with the zero of the second and third derivative after the maximum temperature of liquidus to obtain a more accurate rigidity temperature point.
We can observe in Figure 5 the determination of the RPT for the third proposed method.”
Point 4: And a brief explanation of why the new methods have its advantage should be presented.
Response 4: In line number 266 new table has been introduced which explain the characteristics and disadvantages of every method statically.
In line 269 a new text has been added:
“Method 1 and Method 2 showed similar coefficient of determination values than Method 3 and Method 4, with values between 0.77-0.78. Method 5 using plotting the derivatives vs temperature parameter is the one with more precision, with 0.82. The use of higher order derivatives that made easier the determination of the RPT and the plotting of the dT/dt curve vs. T, which it is independent of the sand cup size provides better results in Method 5. A value of the coefficient of determination between 0.5 and 0.8 is classified as a regular and a value from 0.8 to 0.9 is classified as a good correlation. We can observe that Method 5 is the only one characterized as good correlation, with the rest of methods near to obtain this classification.
The standard error is similar for Method 4 and 5, and slightly higher in Method 1,2 and 3. A difference of about 1.4ºC in the RPT with deviation percentages about 0.4% are acceptable when the approximative RPT temperature is about 565ºC.”
Also in the new figure number 2 we can appreciate why these methods cannot be applied to all type of AlSi10Mg alloys.
Point 5: It would be nice to show the schematic on the experiment setup, it is not so clear whether the experiment setup of the new methods is the same as the commonly used ones.
Response 5: In line number 226 new figure describing the whole procedure followed in this research work has been introduced.
Point 6: In section 2, there is only one subsection 2.1, is there anything missed here?
Response 6: In fact, more than a sub-section, it was intended to explain the different methods. According to suggested the text has been modified according to its corresponding design from line 85 to line 100.
“There are four main processes for the determination of RPT temperature. The first, the mechanical-rheological method, is based on the continuous recording of the torque required to produce the rotation of a disc or paddle into a liquid metal [12]-[13] until the dendrite structure becomes mechanically rigid, stopping the rotation of the impeller. The temperature at which this occurs is defined as the RPT.
The second is based on the in-situ study of neutron diffraction during casting. However, this system needs a large detector to obtain enough resolution of the diffraction peak in the semi-solid state [6].
The third method corresponds to the two thermocouples thermal analysis technique [14] to determine the temperature data at the center (Tc) and at a nearby inner wall (Tw) with two thermocouples. The RPT is determined with the second minimum on the DT versus time curve (DT = Tw -Tc) and its projection on the Tc cooling curve. Due to the difference in the thermal conductivity in the solid and liquid phases, there is a minimum on the DT versus time curve.
There are other methods that are based on employing only one thermocouple to decrease costs and to increase productivity of the data analysis. But there is no work in which this method has been applied to aluminum alloys”
Point 7: Also, Is there any advantage or disadvantage of each of the three methods?
Response 7: According to the suggestion new manuscripts has been completed including complete information as for example in line number 322 where it is explained the advantage of method number 5.
“Method 5 is the most accurate process for determining the RPT values, with the highest coefficient of regression, combined with a small error. In addition to using higher order derivatives, it uses the dT/dt derivative curve with respect to the temperature in order to define the RPT. It’s similar to method 3 in using higher order derivatives to determine the RPT but in this case the temperature derivative curve with respect to the time in a way similar to Method 4, thus the coefficient of regression is reduced. “

Round 2
Reviewer 1 Report
I suggest to re-group the data in Table 1. Consider numbering/listing the samples based first on the content of Si - from lower concentrations to higher concentrations. For equal concentration order the samples according to the content of either Fe of Mg. Do change the Figure 7 accordingly (the numbers assigned to a given composition shall change)
Author Response
Point 1: I suggest to re-group the data in Table 1. Consider numbering/listing the samples based first on the content of Si - from lower concentrations to higher concentrations. For equal concentration order the samples according to the content of either Fe of Mg.
Response 1: According to suggested, table 1 has been re-adapted, listing the samples according to the silicon content from the lowest to the highest wt%. concentration and for identical Si concentrations according to the content of Fe and Mg.
Point 2: Do change the Figure 7 accordingly (the numbers assigned to a given composition shall change)
Response 2: The figure number 7 has been modified applying the same criteria and also in the discussion of results it has been made an analysis of the alloys that show the smaller RPT values, according to the alloying elements wt.%.
